# Unsupervised Video Decomposition using Spatio-temporal Iterative Inference

## Abstract

Unsupervised multi-object scene decomposition is a fast-emerging problem in representation learning. Despite significant progress in static scenes, such models are unable to leverage important dynamic cues present in video. We propose a novel spatio-temporal iterative inference framework that is powerful enough to jointly model complex multi-object representations and explicit temporal dependencies between latent variables across frames. This is achieved by leveraging 2D-LSTM, temporally conditioned inference and generation within the iterative amortized inference for posterior refinement. Our method improves the overall quality of decompositions, encodes information about the objects' dynamics, and can be used to predict trajectories of each object separately. Additionally, we show that our model has a high accuracy even without color information. We demonstrate the decomposition, segmentation, and prediction capabilities of our model and show that it outperforms the state-of-the-art on several benchmark datasets, one of which was curated for this work and will be made publicly available.

## 1 Introduction

Unsupervised representation learning, which has a long history dating back to Boltzman Machines (Hinton & Sejnowski, 1986) and original works of Marr (1970), has recently emerged as one of the important directions of research, carrying the newfound promise of alleviating the need for excessively large and fully labeled datasets. More traditional representation learning approaches focus on unsupervised (*e.g.*, autoencoder-based (Pathak et al., 2016; Vincent et al., 2008)) or self-supervised (Noroozi & Favaro, 2016; Vondrick et al., 2016; Zhang et al., 2016) learning of *holistic* representations that, for example, are tasked with producing (spatial (Noroozi & Favaro, 2016), temporal (Vondrick et al., 2016), or color (Zhang et al., 2016)) encodings of images or patches. The latest and most successful methods along these lines include ViLBERT (Lu et al., 2019) and others (Sun et al., 2019; Tan & Bansal, 2019) that utilize powerful transformer architectures (Vaswani et al., 2017) coupled with proxy multi-modal tasks (*e.g.*, masked token prediction or visua-lingual alignment). Learning of good *disentangled*, spatially *granular*, representations that are, for example, able to decouple object appearance and shape in complex visual scenes consisting of multiple moving objects remains elusive.

Recent works that attempt to address this challenge can be characterized as: (i) attention-based methods (Crawford & Pineau, 2019b; Eslami et al., 2016), which infer latent representations for each object in a scene, and (ii) iterative refinement models (Greff et al., 2019; 2017), which decompose a scene into a collection of components by grouping pixels. Importantly, the former have been limited to latent representations at object- or image patch-levels, while the latter class of models have illustrated the ability for more granular latent representations at the pixel (segmentation)-level. Specifically, most refinement models learn pixel-level generative models driven by spatial mixtures (Greff et al., 2017) and utilize amortized iterative refinements (Marino et al., 2018) for inference of disentangled latent representations within the VAE framework (Kingma & Welling, 2014); a prime example is IODINE (Greff et al., 2019). However, while providing a powerful model and abstraction which is able to segment and disentangle complex scenes, IODINE (Greff et al., 2019) and other similar architectures are fundamentally limited by the fact that they only consider images. Even when applied for inference in video, they process one frame at a time. This makes it excessively challenging to discover and represent individual instances of objects that may share properties such as appearance and shape but differ in dynamics.

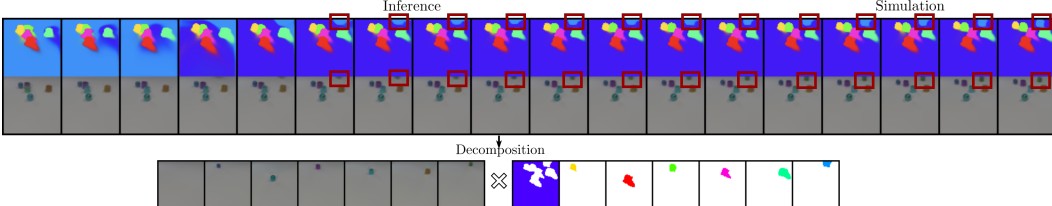

Figure 1: **Unsupervised Video Decomposition.** Our approach allows to infer precise segmentations of the objects via interpretable latent representations, that can be used to decompose each frame and simulate the future dynamics, all in unsupervised fashion. Whenever a new object emerges into a frame the model dynamical adapts and uses one of the segmentation slots to assign to the new object.

In computer vision, it has been a long-held belief that motion carries important information for segmenting objects (Jepson et al., 2002; Weiss & Adelson, 1996). Armed with this intuition, we propose a spatio-temporal amortized inference model capable of not only unsupervised multi-object scene decomposition, but also of learning and leveraging the implicit probabilistic dynamics of each object from perspective raw video alone. This is achieved by introducing temporal dependencies between the latent variables across time. As such, IODINE (Greff et al., 2019) could be considered a special (spatial) case of our spatio-temporal formulation. Modeling temporal dependencies among video frames also allows us to make use of conditional priors (Chung et al., 2015) for variational inference, leading to more accurate and efficient inference results.

The resulting model, illustrated in Fig. 1, achieves superior performance on complex multi-object benchmark datasets (Bouncing Balls and CLEVRER) with respect to state-of-the-art models, including R-NEM (Van Steenkiste et al., 2018) and IODINE (Greff et al., 2019) in terms of segmentation, prediction, and generalization. Our model has a number of appealing properties, including temporal extrapolation, computational efficiency, and the ability to work with complex data exhibiting non-linear dynamics, colors, and changing number of objects within the same video sequence. In addition, we introduce an entropy prior to improve our model's performance in scenarios where object appearance alone is not sufficiently distinctive (*e.g.*, greyscale data).

## 2 RELATED WORK

**Unsupervised Scene Representation Learning.** Unsupervised scene representation learning can generally be divided into two groups: attention-based methods, which infer latent representations for each object in a scene, and more complex and powerful iterative refinement models, which often make use of spatial mixtures and can decompose a scene into a collection of estimated components by grouping pixels together. *Attention-based* methods, such as AIR (Eslami et al., 2016) (Xu et al., 2019) and SPAIR (Crawford & Pineau, 2019b), decompose scenes into latent variables representing the appearance, position, and size of the underlying objects. However, both methods can only infer the objects' bounding boxes and have not been shown to work on non-trivial 3D scenes with perspective distortions and occlusions. MoNet (Burgess et al., 2019) is the first model in this family tackling more complex data and inferring representations that can be used for instance segmentation of objects. On the other hand, it is not a probabilistic generative model and thus not suitable for density estimation. GENESIS (Engelcke et al., 2020) extends it and alleviates some of its limitations by introducing a probabilistic framework and allowing for spatial relations between the objects. DDPAE (Hsieh et al., 2018) is a framework that uses structured probabilistic models to decompose a video into low-dimensional temporal dynamics with the sole purpose of prediction. It is shown to operate on binary scenes with no perspective distortion and is not capable of generating per-object segmentation masks. *Iterative refinement* models started with Tagger (Greff et al., 2016) that reasons about the segmentation of its inputs. However, it does not allow explicit latent representations and cannot be scaled to more complex images. NEM (Greff et al., 2017), as an extension of Tagger, uses a spatial mixture model inside an expectation maximization framework, but is limited to binary data. Finally, IODINE (Greff et al., 2019) is a notable example of a model employing iterative amortized inference w.r.t. a spatial mixture formulation and achieves state-of-the-art performance in scene decomposition and segmentation.

**Unsupervised Video Tracking and Object Detection.** SQAIR (Kosiorek et al., 2018), SILOT (Crawford & Pineau, 2019a) and SCALOR (Jiang et al., 2020) are temporal extensions

of the static attention-based models that are tailored to tracking and object detection tasks. SQAIR is restricted to binary data and operates at the level of bounding boxes. SILOT and SCALOR are more expressive and can cope with cluttered scenes, a larger numbers of objects, and dynamic backgrounds, but do not work on colored perspective[1] data; accurate segmentation remains a challenge.

**Unsupervised Video Decomposition and Segmentation.** Models employing spatial mixtures and iterative inference in a temporal setting are closest to our method from a technical perspective. Notably, there are only few models falling into this line of work: RTagger (Prémont-Schwarz et al., 2017) is a recurrent extension of Tagger and has same limitations as its predecessor. R-NEM (Van Steenkiste et al., 2018) effectively learns the objects' dynamics and interactions through a relational module and can produce segmentations but is limited to orthographic binary data.

**Methods without Latent Modeling.** GAN-based ReDO (Chen et al., 2019) uses a model built around the assumption that object regions are independent, guiding the generator by drawing objects' pixel regions separately and composing them after segmentation. Another model (Arandjelović & Zisserman, 2019) employs the same principles but guide the generator by copying a region of an image into another one. Both architectures are shown to operate on static images only and do not have a clearly interpretable latent space or prediction capabilities.

Our method allows an effective use of temporal information in object-centric decompositions of colored video data. This places our approach between methods like R-NEM, which strictly operates on binary data, and IODINE, whose usage of temporal information is ad-hoc and produces results of limited quality (Table 1). In practice, we leverage a 2D-LSTM and employ an implicit modeling of dynamics by incorporating the hidden states into a conditional prior in the efficient runtime manner.

## 3 DYNAMIC VIDEO DECOMPOSITION

We now introduce our dynamic model for unsupervised video decomposition. Our approach builds upon a generative model of multi-object representations and leverages elements of iterative amortized inference. We briefly review both concepts (§3.1) and then introduce our model (§3.2).

### 3.1 BACKGROUND

**Multi-Object Representations.** The multi-object framework introduced in Greff et al. (2019) decomposes a static image $\mathbf{x} = (x_i)_i \in \mathbb{R}^D$ into $K$ objects (including background). Each object is represented by a latent vector $\mathbf{z}^{(k)} \in \mathbb{R}^M$ capturing the object's unique appearance and can be thought of as an encoding of common visual properties, such as color, shape, position, and size. For each $\mathbf{z}^{(k)}$ independently, a broadcast decoder (Watters et al., 2019) generates pixelwise pairs $(m_i^{(k)}, \mu_i^{(k)})$ describing the assignment probability and appearance of pixel $i$ for object $k$. Together, they induce the generative image formation model

$$p(\mathbf{x}|\mathbf{z}) = \prod_{i=1}^{D} \sum_{k=1}^{K} m_i^{(k)} \mathcal{N}(x_i; \ \mu_i^{(k)}, \sigma^2),\tag{1}$$

where $\mathbf{z} = (\mathbf{z}^{(k)})_k$, $\sum_{k=1}^{K} m_i^{(k)} = 1$ and $\sigma$ is the same and fixed for all $i$ and $k$. The original image pixels can be reconstructed from this probabilistic representation as $\widetilde{x}_i = \sum_{k=1}^{K} m_i^{(k)} \mu_i^{(k)}$.

**Iterative Amortized Inference.** Our approach leverages the iterative amortized inference framework (Marino et al., 2018), which uses the learning to learn principle (Andrychowicz et al., 2016) to close the amortization gap (Cremer et al., 2017) typically observed in traditional variational inference. The need for such an iterative process arises due to the multi-modality of Eq.(1), which results in an order invariance and assignment ambiguity in the approximate posterior that standard variational inference cannot overcome (Greff et al., 2019).

The idea of amortized iterative inference is to start with randomly guessed parameters $\boldsymbol{\lambda}_1^{(k)}$ for the approximate posterior $q_{\boldsymbol{\lambda}}(\mathbf{z}_1^{(k)}|\mathbf{x})$ and update this initial estimate through a series of $R$ refinement steps. Each refinement step $r \in \{1, \dots, R\}$ first samples a latent representation from $q_{\boldsymbol{\lambda}}$ to evaluate

---

[1]Perspective videos are more complex as objects can occlude one another and change in size over time.

the ELBO $\mathcal{L}$ and then uses the approximate posterior gradients $\nabla_{\boldsymbol{\lambda}}\mathcal{L}$ to compute an additive update $f_\phi$, producing a new parameter estimate $\boldsymbol{\lambda}_{r+1}^{(k)}$:

$$\mathbf{z}_r^{(k)} \overset{k}{\sim} q_{\boldsymbol{\lambda}}(\mathbf{z}_r^{(k)}|\mathbf{x}), \qquad\qquad \boldsymbol{\lambda}_{r+1}^{(k)} \overset{k}{\leftarrow} \boldsymbol{\lambda}_r^{(k)} + f_\phi(\mathbf{a}^{(k)}, \mathbf{h}_{r-1}^{(k)}), \qquad\qquad (2)$$

where $\mathbf{a}^{(k)}$ is a function of $\mathbf{z}_r^{(k)}$, $\mathbf{x}$, $\nabla_{\boldsymbol{\lambda}}\mathcal{L}$, and additional inputs (mirrors definition in Greff et al. (2019)). The function $f_\phi$ consists of a sequence of convolutional layers and an LSTM. The memory unit takes as input a hidden state $\mathbf{h}_{r-1}^{(k)}$ from the previous refinement step.

## 3.2 SPATIO-TEMPORAL ITERATIVE INFERENCE

Our proposed model builds upon the concepts introduced in the previous section and enables robust learning of dynamic scenes through spatio-temporal iterative inference. Specifically, we consider the task of decomposing a video sequence $\mathbf{x} = (\mathbf{x}_t)_{t=1}^T = (x_{t,i})_{t,i=1}^{T,D}$ into $K$ slot sequences $(\mathbf{m}_t^{(k)})_t$ and $K$ appearance sequences $(\boldsymbol{\mu}_t^{(k)})_t$. To this end, we introduce explicit temporal dependencies into the sequence of posterior refinements and show how to leverage this contextual information during decoding with a generative model. The resulting computation graph can be thought of as a 2D grid with time dimension $t$ and refinement dimension $r$ (Fig. 2a). Propagation of information along these two axes is achieved with a 2D-LSTM (Graves et al., 2007) (Fig. 2b), which allows us to model the joint probability over the entire video sequence inside the iterative amortized inference framework. The proposed method is expressive enough to model the multimodality of our image formation process and posterior, yet its runtime complexity is smaller than that of its static counterpart.

### 3.2.1 VARIATIONAL OBJECTIVE

Since exact likelihood training is intractable, we formulate our task in terms of a variational objective. In contrast to traditional optimization of the evidence lower bound (ELBO) through static encodings of the approximate posterior, we incorporate information from two dynamic axes: (1) variational estimates from previous refinement steps; (2) temporal information from previous frames. Together, they form the basis for spatio-temporal variational inference via iterative refinements. Specifically, we train our model by maximizing the following ELBO objective[2]:

$$\mathcal{L}_{\text{ELBO}}(\mathbf{x}) = \mathbb{E}_{q_{\boldsymbol{\lambda}}(\mathbf{z}_{\leq T,R}|\mathbf{x}_{\leq T})} \sum_{t=1}^T \sum_{r=1}^{\widehat{R}} \Big[ \beta \log\left(p\left(\mathbf{x}_t|\mathbf{x}_{<t}, \mathbf{z}_{\leq t,r}\right)\right) - \text{KL}(q_{\boldsymbol{\lambda}}(\mathbf{z}_{t,r}|\mathbf{x}_{\leq t}, \mathbf{z}_{<t,r}) \,||\, p(\mathbf{z}_t|\mathbf{x}_{<t}, \mathbf{z}_{<t})) \Big], \quad (3)$$

where the first term expresses the reconstruction error of a single frame and the second term measures the divergence between the variational posterior and the prior. The relative weight between terms is controlled with a hyperparameter $\beta$ (Higgins et al., 2017). Furthermore, to reduce the overall complexity of the model and to make it easier to train, we set $\widehat{R} := \max(R - t, 1)$ (see Fig. 2 for an illustration). Compared to a static model, which infers each frame independently, reusing information from previous refinement steps also makes our model more computationally efficient. In the next sections, we discuss the form of the conditional distributions in Eq.(3) in more detail.

### 3.2.2 INFERENCE AND GENERATION

**Posterior Refinement.** Optimizing Eq.(3) inside the iterative amortized inference framework (Section 3.1) requires careful thought about the nature and processing of the hidden states. While there is vast literature on the propagation of a single signal, including different types of RNNs (Hochreiter & Schmidhuber, 1997; Cho et al., 2014; Graves et al., 2005; Chung et al., 2017) and transformers (Vaswani et al., 2017), the optimal solution for multiple axes with different semantic meaning (*i.e.*, time and refinements) is less obvious. Here, we propose to use a 2D version of the uni-directional MD-LSTM (Graves et al., 2007) to compute our variational objective (Eq.(3)) in an iterative manner. In order to do so, we replace the traditional LSTM in the refinement network (Eq.(2)) with a 2D extension. This extension allows the posterior gradients to flow through both the grid of the previous refinements and the previous time steps (see Fig. 2a). Writing $\mathbf{z}_{t,r}$ for the latent encoding at time $t$ and refinement $r$, we can formalize this new update scheme as follows:

$$\mathbf{z}_{t,r} \sim q_{\boldsymbol{\lambda}}(\mathbf{z}_{t,r}|\mathbf{x}_{\leq t}, \mathbf{z}_{<t,r}), \qquad\qquad \boldsymbol{\lambda}_{t,r+1} \leftarrow \boldsymbol{\lambda}_{t,r} + f_\phi(\mathbf{a}, \mathbf{h}_{t,r-1}, \mathbf{h}_{t-1,\widehat{R}}). \qquad (4)$$

---

[2]For simplicity, we drop references to the object slot $\bullet^{(k)}$ from now on and formulate all equations on a per-slot basis.

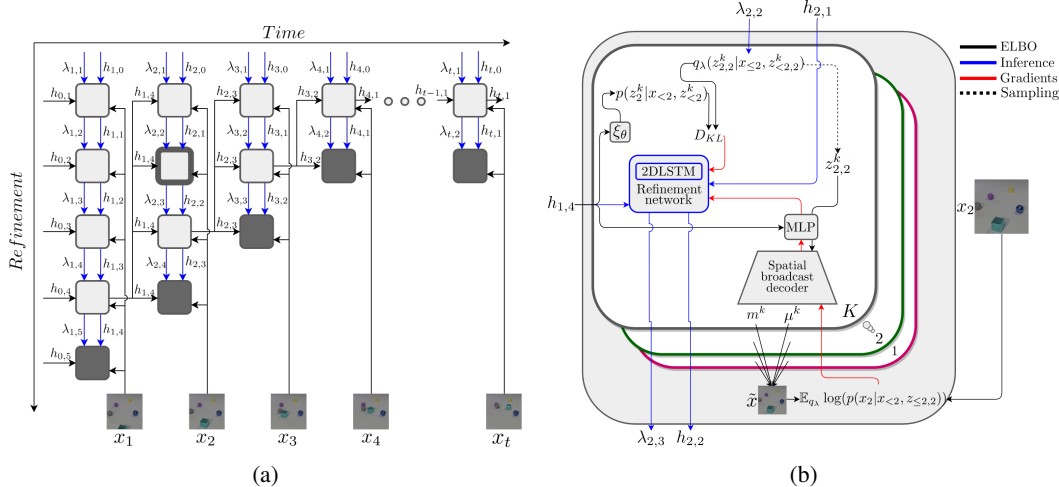

(a)                                                              (b)

Figure 2: **Model Overview.** (**a**) Inference in our model passes through a 2D grid in which light gray cell $(r, t)$ represents the $r$-th refinement at time $t$, dark gray cells are where the final reconstruction is computed and no refinement is needed . Each light gray cell receives three inputs: a refinement hidden state $\mathbf{h}_{t,r-1}$, a temporal hidden state $\mathbf{h}_{t-1,\widehat{R}}$, and posterior parameters $\boldsymbol{\lambda}_{t,r}$. The outputs are a new hidden state $\mathbf{h}_{t,r}$ and new posterior parameters $\boldsymbol{\lambda}_{t,r+1}$. (**b**) An example of the internal structure of the highlighted cell from Fig. (a). We process the inputs with the help of a spatial broadcast decoder and a 2D LSTM. The rest of the light gray cells have the same structure.

Note that the hidden state from the previous time step is always $\mathbf{h}_{t-1,\widehat{R}}$, *i.e.*, the one computed during the final refinement $\widehat{R}$ at time $t-1$. Our reasoning for this is that the approximation of the posterior only improves with the number of refinements (Marino et al., 2018).

**Temporal Conditioning.** Inside the learning objective we set the prior and the likelihood to be conditioned on the previous frames and the refinement steps. This naturally comes from an idea that each frame is dependent on the predecessor's dynamics and therefore latent representations should follow the same property. Conditioning on the refinement steps is essential to the iterative amortized inference procedure. To model the prior and the likelihood distributions accordingly we adopt the approach proposed in Chung et al. (2015) but tailor it to our iterative amortized inference setting. Specifically, the parameters of our Gaussian prior are now computed from the temporal hidden state $\mathbf{h}_{t-1,\widehat{R}}$:

$$p(\mathbf{z}_t|\mathbf{x}_{<t}, \mathbf{z}_{<t}) = \mathcal{N}(\mathbf{z}_t; \widetilde{\boldsymbol{\mu}}_t, \operatorname{diag}(\widetilde{\boldsymbol{\sigma}}_t^2)), \quad [\widetilde{\boldsymbol{\mu}}_t, \widetilde{\boldsymbol{\sigma}}_t] = \xi_\theta(\mathbf{h}_{t-1,\widehat{R}}), \tag{5}$$

where $\xi_\theta$ is a simple neural network with a few layers.[3] Please refer to the supplemental material for details. Note that the prior only changes along the time dimension and is independent of the refinement iterations, because we refine the posterior to be as close as possible to the dynamic prior for the current time step. Finally, to complete the conditional generation, we modify the likelihood distribution as follows[4]:

$$p(\mathbf{x}_t|\mathbf{x}_{<t}, \mathbf{z}_{\le t,r}) = \prod_{i=1}^{D} \sum_{k=1}^{K} m_{t,r,i}^{(k)} \mathcal{N}(x_{t,i}; \mu_{t,r,i}^{(k)}, \sigma^2), \quad [m_{t,r,i}^{(k)}, \mu_{t,r,i}^{(k)}] = g_\theta(\mathbf{z}_{t,r}^{(k)}, \mathbf{h}_{t-1,\widehat{R}}^{(k)}), \tag{6}$$

where $\mu_{t,r,i}^{(k)}, m_{t,r,i}^{(k)}$ are mask and appearance of pixel $i$ in slot $k$ at time step $t$ and refinement step $r$. $g_\theta$ is a spatial mixture broadcast decoder (Greff et al., 2019) with preceding MLP to transform the pair $\left( \mathbf{z}_{t,r}^{(k)}, \mathbf{h}_{t-1,\widehat{R}}^{(k)} \right)$ into a single vector representation.

### 3.2.3 LEARNING AND PREDICTION

**Architecture.** From a graphical point of view, we can think of the refinement steps and time steps as being organized on a 2D grid from Fig. 2a, with light gray cell $(r, t)$ representing the $r$-th refinement at

---

[3]In practice, $\xi_\theta$ predicts $\log \boldsymbol{\sigma}_t$ for stability reasons.

[4]Since our likelihood is a Gaussian mixture model, we are now referencing the object slot $\bullet^{(k)}$ again.

time $t$. According to Eq.(4), each such cell takes as input the hidden state from a previous refinement $\mathbf{h}_{t,r-1}$, the temporal hidden state $\mathbf{h}_{t-1,\widehat{R}}$, and the posterior parameters $\boldsymbol{\lambda}_{t,r}$. Outputs of each light gray cell are new posterior parameters $\boldsymbol{\lambda}_{t,r+1}$ and a new hidden state $\mathbf{h}_{t,r}$. At the last refinement $\widehat{R}$ at time $t$, the value of the refinement hidden state $\mathbf{h}_{t,r}$ is assigned to a new temporal hidden state $\mathbf{h}_{t,\widehat{R}}$.

**Training Objective.** Instead of a direct optimization of Eq.(3), we propose two modifications that we found to improve our model's practical performance: (1) similar to observations made by (Greff et al., 2019), we found that color is an important factor for high-quality segmentations. In the absence of such information, we mitigate the arising ambiguity by maximizing the entropy of the masks $m_{t,r,i}^{(k)}$ along the slot dimension $k$, *i.e.*, we train our model by maximizing the objective

$$\mathcal{L}_{\text{ELBO}} + \gamma \sum_{i=1}^{D} \sum_{k=1}^{K} m_{t,r,i}^{(k)} \log(m_{t,r,i}^{(k)}), \tag{7}$$

where $\gamma$ defines the weight of the entropy loss. (2) In addition to the entropy loss, we also prioritize later refinement steps by weighting the terms in the inner sum of Eq.(3) with $\frac{r}{R}$.

**Prediction.** On top of pure video decomposition, our model is also able to simulate future frames $\mathbf{x}_{T+1}, \ldots, \mathbf{x}_{T+T'}$. Because our model requires image data $\mathbf{x}_t$ as input, which is not available during simulation of new frames, we use the reconstructed image $\widetilde{\mathbf{x}}_t$ in place of $\mathbf{x}_t$ to compute the likelihood $p(\mathbf{x}_t | \mathbf{x}_{<t}, \mathbf{z}_{\leq t,r})$ in these cases. We also set the gradients $\nabla_{\boldsymbol{\lambda}}\mathcal{L}$, $\nabla_{\boldsymbol{\mu}}\mathcal{L}$, and $\nabla_{\mathbf{m}}\mathcal{L}$ to zero.

**Complexity.** Our model's ability to reuse information from previous refinements leads to a runtime complexity of $\mathcal{O}(R^2 + T)$, which is much more efficient than the $\mathcal{O}(RT)$ complexity of the traditional IODINE model (when each frame is inferred independently) in the typical case of $T \gg R$.

## 4 EXPERIMENTS

We validate our model on Bouncing Balls (Van Steenkiste et al., 2018), an augmented version of CLEVRER (Yi et al., 2020), and Grand Central Station (Zhou et al., 2012). Our experiments comprise quantitative studies of decomposition quality during generation and prediction, as well as an ablation study. Also see Appendices C, D and F.

### 4.1 SETUP

**Datasets.** Bouncing Balls consists of 50 frame, binary, $64 \times 64$ resolution video sequences. Each video shows simulated balls with different masses bouncing elastically off each other and the image border. We train our model on the first 40 frames of 50K videos containing 4 balls in each frame. We use two different test sets consisting of 10K videos with 4 balls and 10K videos with 6-8 balls. We also validate our model on a color version of this dataset that we generate using the segmentation masks.

CLEVRER contains synthetic videos of moving and colliding objects. Each video is 5 seconds long (128 frames) at resolution $480 \times 320$, which we trim and rescale to $64 \times 64^5$ pixels (see Appendix B). For training, we use the same 10K videos as in the original source. For testing, we compute ground truth masks for the validation set using the provided annotations and test on 2.5K instances containing 3-5 objects and on 1.1K instances containing 6 objects. In the training, we set the number of slots $K$ to 6 for CLEVRER and to one more than the maximum number of objects in all other cases.

Grand Central Station is a video feed from the main hall of a trafficked train station, containing a high number of people moving at various paces in all different directions. It has a total of 50010 frames in a resolution of $720 \times 480$. In order to make the dataset more manageable, we have extracted a portion of the feed of resolution $128 \times 128$ and segmented it into sequences of 20 frames each. Each sequence contains approximately 10 people. We set $K$ to 8 during training and to 10 for testing.

**Baselines.** We compare our approach to recent baselines: R-NEM (Van Steenkiste et al., 2018), IODINE (Greff et al., 2019) and DDPAE (Hsieh et al., 2018). R-NEM is a state-of-the-art model for unsupervised video decomposition and physics learning. While showing impressive results on simulation tasks, it is limited to binary data and has difficulties with perspective scenes. IODINE is more expressive but static in nature and cannot capture temporal dynamics within its probabilistic

---

[5]The method is robust enough to handle 128x128 resolution as it is built on top of the IODINE.

Table 1: **Quantitative Evaluation (Scene Decomposition).** We show our model's ability to produce high-quality instance segmentations for sequences with varying length. We test on sequences with 4 balls and two different types of data (binary, colored) for Bouncing Balls and on sequences with 3-5 objects for CLEVRER. Note, R-NEM does not cope with color data; hence we only run it on binary.

| | | ARI (↑) | | | | F-ARI (↑) | | | | MSE (↓) | | | |
|---|---|---|---|---|---|---|---|---|---|---|---|---|---|
| | **Bouncing Balls** | | | | | | | | | | | | |
| | Length | 10 | 20 | 30 | 40 | 10 | 20 | 30 | 40 | 10 | 20 | 30 | 40 |
| binary | R-NEM | 0.5031 | 0.6199 | 0.6632 | 0.6833 | 0.6259 | 0.7325 | 0.7708 | 0.7899 | 0.0252 | 0.0138 | 0.0096 | 0.0076 |
| | IODINE | 0.0318 | | | | 0.9986 | | | | 0.0018 | | | |
| | SEQ-IODINE | 0.0230 | 0.0223 | 0.0021 | -0.0201 | 0.8645 | 0.6028 | 0.5444 | 0.4063 | 0.0385 | 0.0782 | 0.0846 | 0.0968 |
| | Our | **0.7169** | **0.7263** | **0.7286** | **0.7294** | **0.9999** | **0.9999** | **0.9999** | **0.9999** | **0.0004** | **0.0004** | **0.0004** | **0.0004** |
| color | IODINE | 0.5841 | | | | 0.9752 | | | | 0.0014 | | | |
| | SEQ-IODINE | 0.3789 | 0.3743 | 0.3225 | 0.2654 | 0.7517 | 0.8159 | 0.7537 | 0.6734 | 0.0160 | 0.0164 | 0.0217 | 0.0270 |
| | Our | **0.7275** | **0.7291** | **0.7298** | **0.7301** | **1.0000** | **1.0000** | **0.9999** | **0.9999** | **0.0002** | **0.0002** | **0.0002** | **0.0002** |
| | **CLEVRER** | | | | | | | | | | | | |
| | | ARI (↑) | | | | F-ARI (↑) | | | | MSE (↓) | | | |
| | Length | 10 | 20 | 30 | 40 | 10 | 20 | 30 | 40 | 10 | 20 | 30 | 40 |
| color | IODINE | 0.1791 | | | | **0.9316** | | | | 0.0004 | | | |
| | SEQ-IODINE | 0.1171 | 0.1378 | 0.1558 | 0.1684 | 0.8520 | 0.8774 | 0.8780 | 0.8759 | 0.0009 | 0.0009 | 0.0010 | 0.0010 |
| | Our | **0.2220** | **0.2403** | **0.2555** | **0.2681** | 0.9182 | 0.9258 | 0.9309 | 0.9312 | **0.0003** | **0.0003** | **0.0003** | **0.0003** |

framework. However, as noted in Greff et al. (2019), it can be readily applied to temporal sequences by feeding a new video frame to each iteration of the LSTM in the refinement network. We call this variant SEQ-IODINE. Since we can perform simulation of short sequences, we include a comparison of the predictive power of our model against DDPAE (Hsieh et al., 2018).

## 4.2 EVALUATION METRICS

**ARI.** The Adjusted Rand Index (Rand, 1971; Hubert & Arabie, 1985) is a measure of clustering similarity. It is computed by counting all pairs of samples that are assigned to the same or different clusters in the predicted and true clusterings. It ranges from -1 to 1, with score of 0 indicating a random clustering and 1 indicating a perfect match. We treat each pixel as one sample and its segmentation as the cluster assignment.

**F-ARI.** The Foreground Adjusted Rand Index is a modification of the ARI score ignoring background pixels, which often occupy the majority of the image. We argue that both metrics are necessary to assess the segmentation quality of a video decomposition method; this metric is also used in (Greff et al., 2019; Van Steenkiste et al., 2018).

**MSE.** The mean squared error between pixels of the reconstructed $\hat{x}$ and the ground truth frames $x$.

## 4.3 VIDEO DECOMPOSITION

We evaluate the models on a video decomposition task at different sequence lengths. As shown in Table 1, our model outperforms the baselines regardless of the presence of color information, which further reduces the error. Our model performs at least 7% better than R-NEM on all metrics and at least 20% than IODINE on ARI and MSE. Since R-NEM cannot cope well with colored data or the perspective of scenes, it is only evaluated on the Bouncing Balls dataset (binary), producing high-error results in the first frames, a phenomenon not observed with our model. IODINE is not designed to utilize temporal information. On both datasets, IODINE's results are therefore computed independently on each frame of the longest sequence. By processing frames separately, IODINE does not keep the same object-slot assignment, which we ignore when computing the scores. SEQ-IODINE tends to perform even worse than IODINE in many experiments, which we attributed to exploding gradients caused by limited refinement steps and a lack of dynamics modeling.

## 4.4 GENERALIZATION

We investigated how well our model adapts to a higher number of objects, evaluating its performance on the Bouncing Balls dataset (6 to 8 objects) and on the CLEVRER dataset (6 objects). Table 2 shows that our F-ARI and MSE scores are at least 50% better than those for R-NEM, and ARI scores are just marginally worse and only on the binary data. In comparison to IODINE we are at least 4% better across all metrics. For the Bouncing Balls dataset we have also investigated the impact of

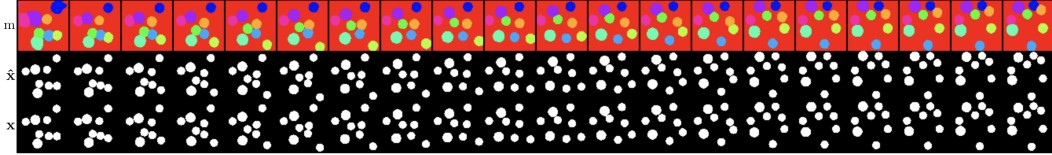

Figure 3: **Qualitative Evaluation (Bouncing Balls).** Our model can generalize to sequences with 8 balls when trained on 4 balls. Top-to-bottom: output masks, reconstructions, and ground truth video.

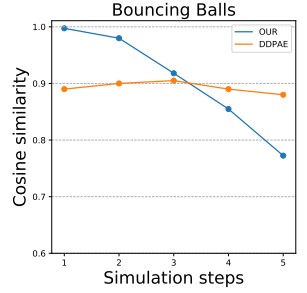

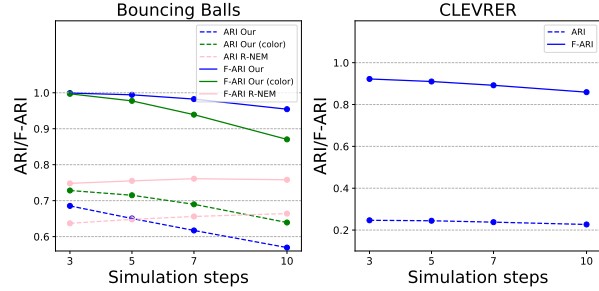

Figure 4: **Velocity predictions.** Cosine similarity for 5 simulated frames after 10 inference steps.

Figure 5: **Prediction.** We show the (F-)ARI for 3, 5, 7, and 10 simulated frames after 20 inference steps.

changing the total number of possible colors to 4 and 8. The former resulting in duplicate colors for different objects and the latter in unique colors for each object. The higher MSE scores for the 8 balls variant is due to the model not being able to reconstruct the unseen colors. Sample qualitative results are shown in Fig. 3 and 6, while more can be found in Appendix F.

### 4.5 PREDICTION

We compare the predictions of our model (Section 3.2.3) to those of R-NEM after 20 steps of inference on 10 predicted steps on the Bouncing Balls dataset (Fig. 5 left). As we can see from the results our model is superior to R-NEM on a shorter sequences, however for the longer sequences we are outperforming R-NEM only on colored data. Our model is capable of more accurate frame prediction than R-NEM on the Bouncing Balls dataset during the first few predicted frames (5-7), with predictions slowly diverging over time due to the temporal consistency. This behavior is also observable on the CLEVRER dataset (Fig. 5 right), albeit to a lesser extent, likely because the objects dynamics are less, even if non-linear. In Figure 4 we compute velocity vectors between bounding box centroids and compare the cosine similarity to the predictions of DDPAE on the Bouncing balls dataset. As expected, our model outperforms DDPAE on the first three frames and then declines in quality. This behavior is not surprising and in line with the results reported in Fig.5

### 4.6 ABLATION

The quantitative results of an ablation study on the binary Bouncing Balls dataset and CLEVRER are shown in Table 3. We investigate the effects of the efficient grid, conditional prior and generation, length of training sequences and entropy term on the performance of our model; all contributions are necessary and important. Note that the base models are too large to be trained on 40 frames, which confirms the superiority of our model in terms of both runtime and memory. The CLEVRER dataset is not binary, which is why we do not include the entropy term (see Section 3.2.3). We validate our choice of $\widehat{R}$ and compare it to alternative options in a supplemental study discussed in Appendix F.2.

## 5 CONCLUSION AND DISCUSSION

We presented a novel unsupervised learning framework capable of precise scene decomposition in multi-object videos with complex appearance and motion. Our temporal component enables modeling of dynamics inside the amortized iterative inference framework but also improves the quality of the results overall. From our quantitative and qualitative comparisons with IODINE and SEQ-IODINE, we see that our model shows more accurate results on the decomposition task. We can detect new

Table 2: **Generalization**. At test time, we change the number of slots in the models from 5 to 9 for the Bouncing Balls test dataset (6-8 balls), and from 6 to 7 for the CLEVRER test dataset (6 objects).

| | | Bouncing Balls | | |
|---|---|---|---|---|
| | | ARI (↑) | F-ARI (↑) | MSE (↓) |
| binary | R-NEM | **0.4484** | 0.6377 | 0.0328 |
| | IODINE | 0.0271 | 0.9969 | 0.0040 |
| | SEQ-IODINE | 0.0263 | 0.8874 | 0.0521 |
| | Our | 0.4453 | **0.9999** | **0.0008** |
| color | IODINE (4) | 0.4136 | 0.8211 | 0.0138 |
| | IODINE (8) | 0.2823 | 0.7197 | 0.0281 |
| | SEQ-IODINE (4) | 0.2068 | 0.5854 | 0.0338 |
| | SEQ-IODINE (8) | 0.1571 | 0.5231 | 0.0433 |
| | Our (4) | 0.4275 | **0.9998** | **0.0004** |
| | Our (8) | **0.4317** | 0.9900 | 0.0114 |

| | | CLEVRER | | |
|---|---|---|---|---|
| | | ARI (↑) | F-ARI (↑) | MSE (↓) |
| color | IODINE | 0.2205 | 0.9305 | 0.0006 |
| | SEQ-IODINE | 0.1482 | 0.8645 | 0.0012 |
| | Our | **0.2839** | **0.9355** | **0.0004** |

Table 3: **Ablation Study**. A 2D-LSTM extension of IODINE trained on sequences of 20 frames is unstable and its output segmentation lacks precision and consistency. Our efficient version of 2D-LSTM grid (Fig. 2a) and the conditional prior and generation increase both segmentation and reconstruction quality. By training these models on longer sequences of 40 frames we observe further improvements.

| | Base | Grid | CP+G | Entropy | Length | ARI (↑) | F-ARI (↑) | MSE (↓) |
|---|---|---|---|---|---|---|---|---|
| BB | ✓ | | | | 20 | 0.0126 | 0.7765 | 0.0340 |
| | ✓ | ✓ | ✓ | | 20 | 0.2994 | 0.9999 | 0.0010 |
| | ✓ | ✓ | ✓ | | 40 | 0.3528 | 0.9998 | 0.0010 |
| | ✓ | ✓ | ✓ | ✓ | 40 | **0.7263** | **0.9999** | **0.0004** |
| CLEVRER | ✓ | | | | 20 | 0.1900 | 0.8200 | 0.0011 |
| | ✓ | ✓ | | | 20 | 0.1100 | 0.9000 | 0.0005 |
| | ✓ | ✓ | ✓ | | 20 | 0.2403 | 0.9258 | **0.0003** |
| | ✓ | ✓ | | | 40 | 0.1700 | 0.9100 | 0.0005 |
| | ✓ | ✓ | ✓ | | 40 | **0.2681** | **0.9312** | **0.0003** |

[Base: base model using 2D-LSTM; Grid: efficient triangular grid structure (Fig. 2a); CP+G: conditional prior and generation; Length: sequence length; Entropy: entropy term (Eq.(7)]

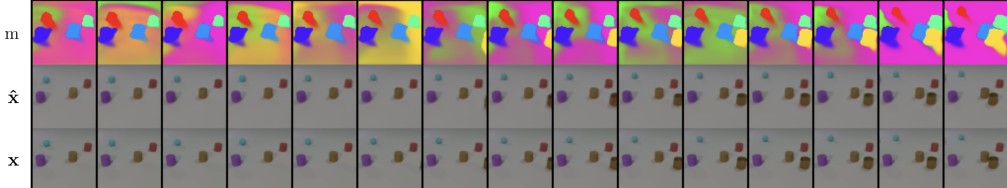

Figure 6: **Qualitative Evaluation (CLEVRER).** Our model can generalize to sequences with 6 objects. We also demonstrate the ability to handle a dynamically changing number of objects, ranging from 4 in the beginning to 6 at the end.

objects faster and are less sensitive to color, because our model can leverage the objects' motion cues. For our experiments, we have chosen a setup consistent with other SOTA methods and a focus on the objects' dynamics. Our model is currently not targeting complex textured datasets, as they are not designed for unsupervised learning and impose additional challenges, such as limited coverage of the input space as well as a superposition of the scene's intrinsic components (object location, articulation, motion, albedo, shading, etc.). We show and discuss the segmentation of a real-world video stream in Fig. 7. Self-supervised methods for video segmentation, which attempt to learn image representations through features, motion or specific tasks, are an alternative but not capable of inferring disentangled representations or extracting interpretable features (e.g. appearance, color). They are typically also not robust to object occlusions, object (dis)appearances, and object ordering (Fig. 14). We refer to Appendix E for an extended discussion and future work.

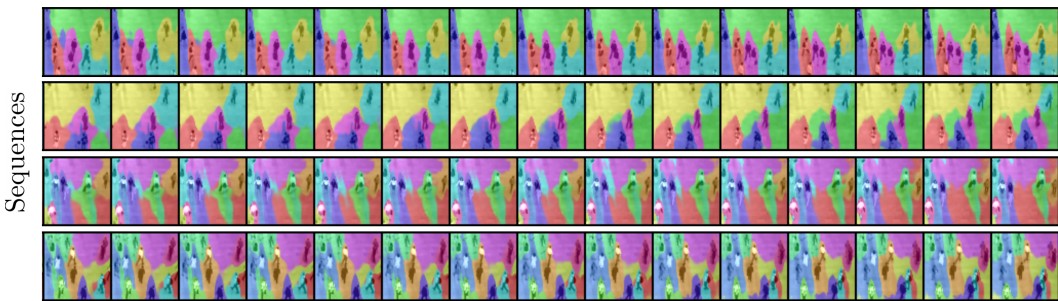

Figure 7: **Qualitative evaluation on real-world data.** Qualitative Evaluation (Grand Central Station). We can observe that our method is very consistent in separating the image regions belonging to different objects as they move in the scene. This dataset is particularly challenging for its background texture, complex lighting and shadows. Please zoom in to allow better clarity.

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

## A    BASELINES

### A.1    R-NEM

We use the R-NEM (Van Steenkiste et al., 2018) authors' original implementation and their publicly available models: `https://github.com/sjoerdvansteenkiste/Relational-NEM`.

### A.2    IODINE

Our IODINE experiments are based on the following PyTorch implementation: `https://github.com/MichaelKevinKelly/IODINE`. We use the same parameters as in this code, with the exceptions of $\beta = 10$ (weight factor) and, for the Bouncing Balls experiments, $R = 6$ (refinement steps). The majority of the hyperparameters shared between our own model and IODINE are identical.

### A.3    SEQ-IODINE

In order to test the sequential version of IODINE, we use the regularly trained IODINE model but change the number of refinement steps to the number of video frames during testing. During each refinement step, instead of computing the error between the reconstructed image and the ground truth image, we use the next video frame. Since the IODINE model was trained on $R = 6$ refinement steps, extending the number of refinement steps to the video length leads to exploding gradients. This effect is especially problematic in the binary Bouncing Balls dataset with 20, 30 and 40 frames per video, because the scores of the static model are already low. We deal with this issue by clamping with max $= 10$ and min $= -10$ the gradients and the $\delta$ refinement value in this experiment[6]. SEQ-IODINE's weak performance, especially w.r.t. the ARI, reflect the gradual divergence from the optimum as the number of frames increases.

## B    DATASETS

**Bouncing Balls.** Bouncing Balls is a dataset provided by the authors of R-NEM (Van Steenkiste et al., 2018).Dataset contains balls with different masses corresponding to their radii. The balls are

---

[6]Please note that clamping was done only when applied to binary Bouncing Balls for 20, 30 and 40 frames.

initialized with random initial positions, masses and velocities. Balls bounce elastically against each other(without occlusions) and the image window. We use the train and test splits of this dataset in two different versions: binary and color. For the color version, we randomly choose 4 colors for the 4-balls (sub-)dataset. For the 6-8 balls test data, we color them in 2 different ways: 4 colors (same as train) and 8 colors (4 from train, 4 new ones). Note that the former results in identical colors for multiple objects, while the latter guarantees unique colors for each object.

**CLEVRER.** Each video in the CLEVRER dataset contains at least one collision and (dis)appearance event making occlusions possible and frequent. Objects' initial velocities are approximately $\pm 2.5\ m/s^2$. Each object has one of eight distinct colors and one of 38 two materials (metal or rubber). In addition, two objects can have the same color but different material.

The version of the CLEVRER dataset (Yi et al., 2020) used in this work was processed as follows:

- Train split, validation split and validation annotations were obtained from the official website: `http://clevrer.csail.mit.edu/`. We use the validation set as test set, because the test set does not contain annotations.

- For training, we use the original train split. Our minimal preprocessing consists of cropping the frames along the width axis by 40 pixels on both sides, followed by a uniform downscaling to 64x64 pixels. Since the length of each video is 128 frames and the maximum number of frames during training was 40, we split the videos into multiple sequences to obtain a larger number of training samples.

- For testing, we trim the videos to a subsequence containing at least 3 objects and object motion. We compute these subsequences by running the script (slice_videos_from_annotations.py in the attached code) from the folder with the validation split and validation annotations.

- The test set ground truth masks can be downloaded from here. The masks and the preprocessed test videos will be grouped into separate folders based on the number of objects in a video.

## C  HYPERPARAMETERS

**Initialization.** We initialize the parameters of the posterior $\boldsymbol{\lambda}$ by sampling from $\mathcal{U}(-0.5, 0.5)$. In all experiments, we use a latent dimensionality $\dim(\boldsymbol{z}) = 64$, such that $\dim(\boldsymbol{\lambda}) = 128$. Horizontal and vertical hidden states and cell states are of size 128, initialized with zeros. $q_{\boldsymbol{\lambda}}$ is the posterior probability per slot of the likelihood $p(\mathbf{x}|\mathbf{z})$, which is a Gaussian mixture model. The variance of the likelihood is set to $\sigma = 0.3$ in all experiments.

**Experiments on Bouncing Balls.** For this experiment, we have explored several values of $R$ (refinement steps) and empirically found $R = 6$ to be optimal in terms of accuracy and efficiency. Refining the posterior more than 6 times does not lead to any substantial improvement, however, the time and memory consumption is significantly increased. For the 4-balls dataset, we use $K = 5$ slots for train and test. For our tests on 6-8 balls, we use $K = 9$ slots. This protocol is identical to the one used in R-NEM (Van Steenkiste et al., 2018). Furthermore, we set $\beta = 100.0$ and scale the KL term by $\psi = 10$. The weight of the entropy term is set to $\gamma = 0.1$ in the binary case. As expected, the effect of the entropy term is most pronounced with binary data, so we set $\gamma = 0$ in all experiments with RGB data.

**Experiments on CLEVRER.** We keep the default number of iterative refinements at $R = 5$, because we did not observe any substantial improvements from a further increase. We use $K = 6$ slots during training, $K = 6$ slot when testing on 3-5 objects and $K = 7$ slots when testing on 6 objects.

## D  TRAINING

We use ADAM (Kingma & Ba, 2014) for all experiments, with a learning rate of 0.0003 and default values for all remaining parameters. During training, we gradually increase the number of frames per video, as we have found this to make the optimisation more stable. We start with sequences of length 4 and train the model until we observe a stagnant loss or posterior collapse. At the beginning

of training, the batch size is 32 and is gradually decreased negatively proportional to the number of frames in the video.

### D.1 Infrastructure and Runtime

We train our models on 8 GeForce GTX 1080 Ti GPUs, which takes approximately one day per model.

### D.2 Code

We have attached the code and the pretrained models to reproduce the experimental results. Please see README file in the code folder to help you with running.

## E Discussion and Future work

Introduction of a temporal component not only enables modelling of dynamics inside the amortized iterative inference framework but also improves the quality of the results overall. From our quantitative and qualitative comparisons with IODINE and SEQ-IODINE, we see that our model shows more accurate results on the decomposition task. We can detect new objects faster and are less sensitive to color, because our model can leverage the objects' motion cues. The ability to work with complex colored data, a property inherited from IODINE, means that we significantly outperform R-NEM. However, R-NEM is a stronger model when it comes to prediction of longer sequences, owing to its ability to model the relations between the objects in the scene. Similar ideas were used in SQAIR (Kosiorek et al., 2018) and GENESIS (Engelcke et al., 2020) by adding a relational RNN (Santoro et al., 2017). Integration of these concepts into our framework is a promising direction for future research. Another possible route is an application of our model to complex real-world scenarios. However, given that such datasets typically contain a much higher number of objects, as well as intricate interactions and spatially varying materials, we consider the resulting scalability questions as a separate line of research.

## F Additional Experiments

### F.1 Prediction

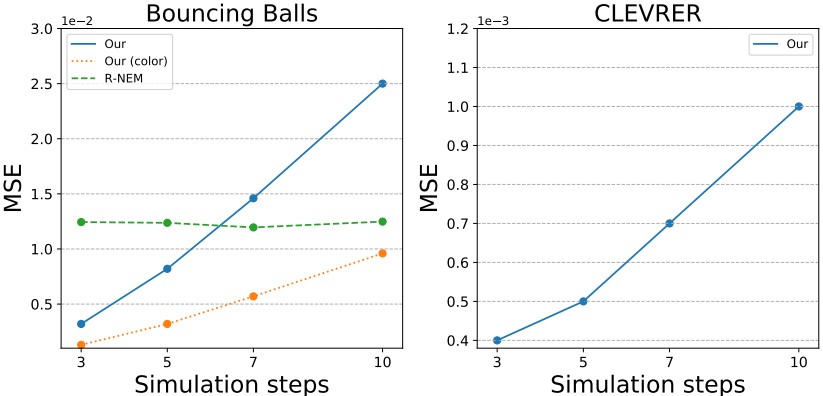

Figure 8: Mean Squared Error for the prediction experiment. We have computed MSE for the same experimental set up as on Fig. 5. As expected the MSE increases with number of simulation steps. Similarly to ARI and F-ARI scores our model outperforms R-NEM on a first steps of simulation, however the error function of our model is grows faster comparatively to R-NEM and we sooner diverge from the accurate simulation.

## F.2 ABLATIONS

The performance of our model is governed by the function $\widehat{R} = \max(R - t, 1)$, where $R$ is the free parameter. In Table 4 we explore values of $R$ ranging from 2 to 10. We see performance saturation at $R \approx 4$. We also explore an alternative choice $\widehat{R}_{\text{alt.}}$ (Table 5), which shows decreased performance compared to $\widehat{R}$. The number of slots $K$ could be determined via cross-validation, but for comparability to other SOTA methods we assume it to be given.

Table 4: **Performance as a Function of Parameter $R$.**

|  | ARI (↑) | | | | F-ARI (↑) | | | | MSE (↓)($\times 10^{-4}$) | | | |
|---|---|---|---|---|---|---|---|---|---|---|---|---|
| R | 2 | 4 | 8 | 10 | 2 | 4 | 8 | 10 | 2 | 4 | 8 | 10 |
| BB bin. | 0.34 | 0.71 | 0.73 | 0.73 | 0.93 | 0.99 | 0.99 | 0.99 | 424 | 6 | 5 | 8 |
| BB col. | 0.48 | 0.72 | 0.73 | 0.73 | 0.93 | 0.99 | 1.0 | 1.0 | 148 | 3 | 3 | 4 |
| CLEVRER | 0.21 | 0.24 | 0.23 | 0.22 | 0.84 | 0.92 | 0.93 | 0.94 | 11 | 3 | 3 | 3 |

Table 5: **Ablation on the Form of the Function $\widehat{R}$.** $\widehat{R}_{\text{alt.}} = R$, when $t = 0$, and $\widehat{R}_{\text{alt.}} = 1$, when $t > 0$.

|  | ARI (↑) | | F-ARI (↑) | | MSE (↓)($\times 10^{-4}$) | |
|---|---|---|---|---|---|---|
|  | $\widehat{R}$ | $\widehat{R}_{alt}$ | $\widehat{R}$ | $\widehat{R}_{alt}$ | $\widehat{R}$ | $\widehat{R}_{alt}$ |
| BB bin. | **0.73** | 0.43 | **1.0** | 0.95 | **4** | 33.2 |
| BB col. | **0.73** | 0.57 | **1.0** | 0.97 | **2** | 11.9 |
| CLEVRER | **0.24** | 0.21 | **0.93** | 0.88 | **3** | 9 |

## F.3 ADDITIONAL QUALITATIVE RESULTS

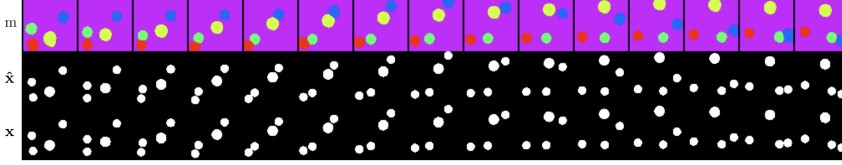

Figure 9: Video decomposition using our model applied on Bouncing Balls dataset with 4 balls.

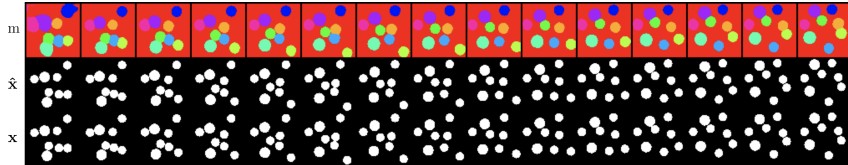

Figure 10: Video decomposition using our model applied on Bouncing Balls dataset with 6-8 balls.

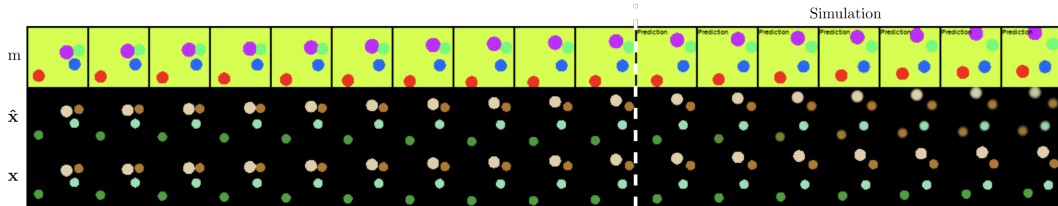

Figure 11: Prediction on Bouncing Balls (colored) dataset.

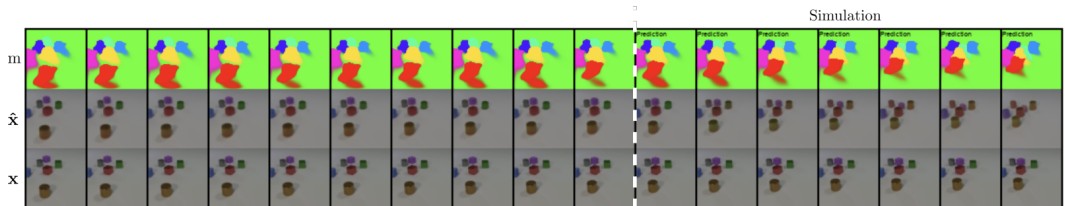

Figure 12: Prediction on CLEVRER dataset.

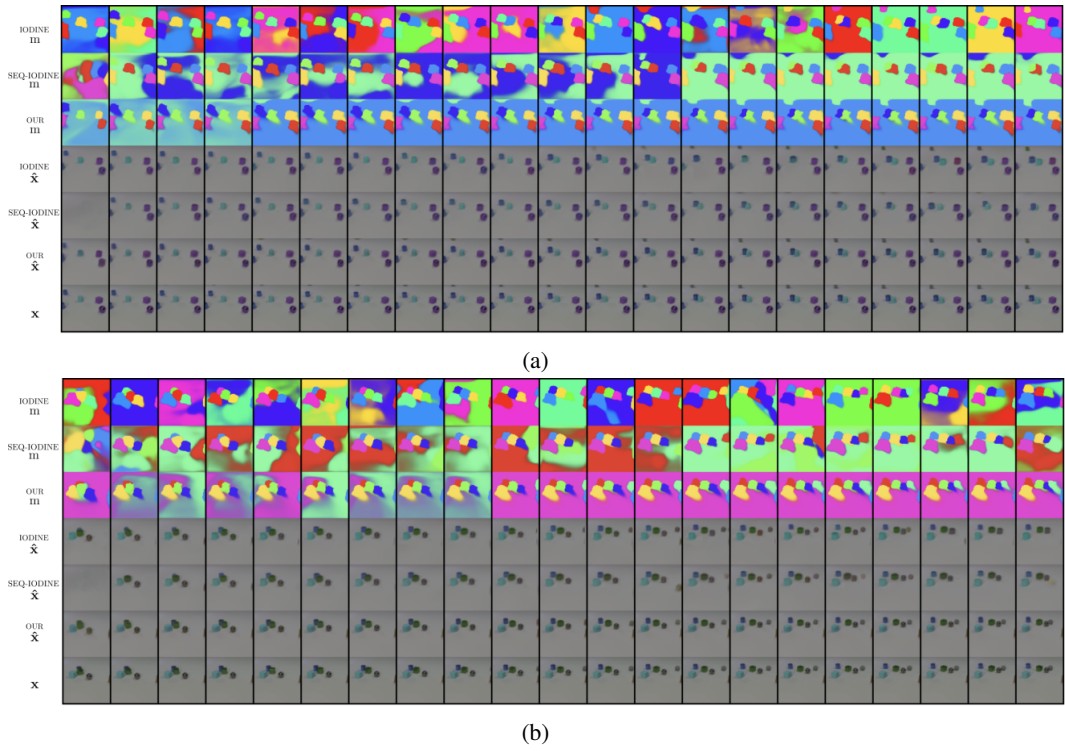

Figure 13: Qualitative results for **Ours vs. IODINE vs. SEQ-IODINE** decomposition experiment. (a) From the figure it is clear that our model can much sooner detect new objects emerging to the frame, while SEQ-IODINE struggles to properly reconstruct and decompose them. And IODINE doesn't have any temporal consistence and reshuffles the slot order. (b). Here we can see that our model is much more stable with time and it does not fail to detect objects, unlike IODINE and SEQ-IODINE.

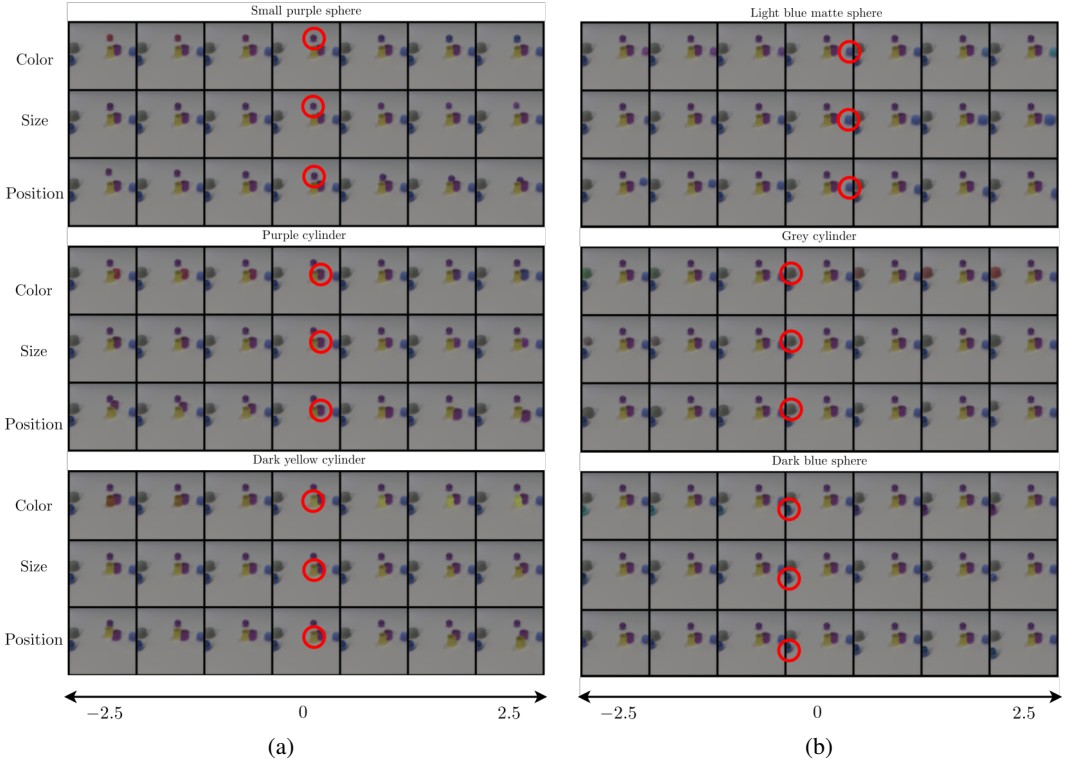

Figure 14: Disentanglement of the latent representations corresponding to distinct interpretable features. CLEVRER latent walks along three different dimensions: color, size and position. We chose a random frame and for each object's representation in the scene dimensions were traversed independently.

## F.4 DISENTANGLEMENT

We demonstrate that introducing a new temporal hidden state and an additional MLP in front of the spatial broadcast decoder has not impacted its ability to separate each object's representations and disentangles them based on color, position, size and other features, similar to results shown in Greff et al. (2019).

## F.5 ANIMATIONS

Attached animations include the following files:

- **bb_binary_4_balls.gif** Animation of the segmentations of 4 binary Bouncing Balls. 50 frames. Here and everywhere else, unless explicitly specified, we also included full scene decomposition and each object's individual reconstruction.

- **bb_binary_6_8_balls.gif** Animation of the ability to generalize to 6-8 binary Bouncing Balls. 40 frames.

- **bb_colored_4_balls.gif** Animation of the 4 colored Bouncing Balls. 50 frames.

- **bb_colored_6_8_balls.gif** Animation of the ability to generalize to 6-8 colored Bouncing Balls. 40 frames.

- **bb_colored_predict.gif** Prediction on the Bouncing Balls colored data. With 40 normal steps of inference and 10 predicted masks and frames. Here we only included predicted masks and ground truth masks.

- **clevrer_5obj.gif** Animation of the segmentations of 5 objects CLEVRER dataset. 50 frames.

- **clevrer_6obj.gif** Animation of the ability to generalize to 6 objects CLEVRER dataset. 45 frames.

