# OpenReview forum: "Unsupervised Video Decomposition using Spatio-temporal Iterative Inference"
_ICLR.cc/2021/Conference — Reject_

### Official Review · AnonReviewer4 · 2020-10-20
**Appropriate evaluation is missing**

**Rating:** 4
**Confidence:** 3

**Review:**

The authors extend previous work of Greff et al. on unsupervised, multi-object scene decomposition to incorporate temporal information.  In particular, they apply the LSTM defined for each candidate object not only over inference steps, but also over time. This allows the model to capture temporal cues, such as object motion, to better decompose the scene into objects. In addition, this allows to speed up the inference, since they perform fewer inference steps at each consecutive frame, capitalizing on temporal consistency in videos (LSTM state can be largely refused between consecutive frames, since the appearance doesn't change a lot). Experimentally demonstrate that their approach indeed outperforms prior work on two toy datasets (bouncing balls and CLEVERER), while being more computationally efficient.

The paper is relatively well written, although it is very dense and is hard to follow for someone not already familiar with the field.

I'm not an expert in iterative amortized inferences, so can't properly judge the technical contribution of the paper. That said, the novelty is non-trivial to the best of my knowledge.

Experimental evaluation is sufficient to compare to the closely related approaches Greff et al., and  Van Steenkiste et al., demonstrating that the proposed method outperforms them when evaluation setting (number of objects in a video) matches the training setting. In a generalization experiments, however, when the number of objects in increased at test time, the gap between the methods becomes much smaller. Moreover, the proposed method seems to generalize worse than Van Steenkiste et al. A reasonable ablation study is provided to justify the design decisions made by the authors.

My main issue is with the positioning of the paper. The authors claim that their goal is learning disentangled, multi-object representations, but all the evaluations are on the task of unsupervised video segmentation on toy datasets. As the authors mention in the conclusion, the problem of unsupervised video segmentation is well studied in the computer vision community, featuring both heuristic-based (Brox and Malik, ECCV'10), and learning-based methods (Xie et al., CVPR'19, Dave et al., ECCV'19 Workshops). Unlike the proposed method, those approaches work on real videos in the wild, though they indeed do not claim to learn an interpretable, disentangled representation. In might well be that this direction of research has merit, even if existing method cannot compete with dedicated video segmentation approaches yet, but this has to be experimentally demonstrated. If video segmentation is indeed not the end goal of your approach, when it shouldn't be used as the main evaluation metric. In particular, when comparing to  Greff et al., and  Van Steenkiste et al., you should demonstrate that the representation learned by your method is more disentangled and more interpretable than theirs, instead of showing that you are better at video segmentation. Indeed, there is no evidence that the two metrics are strongly correlated. In addition, this would allow you to compare your method to other, not object decomposition-based methods for learning disentangled representations.

As mentioned above, I cannot adequately judge the technical contribution of the paper, so, if other reviewers find it convincing in itself, I'm ready to increase my score. The experimental evaluation of the actual task studied in the paper (learning disentangled representations) is completely missing, however, thus I recommend to reject the paper in this round.


Some minor comments:

1. Why not use mask IoU for segmentation evaluation?

2. Why are ARI scores higher on CLEVERER in Table 2 compare to Table 1 (generalization scenario with more objects)?

3. Please discuss and compare to Hsieh et al., NIPS'18, who also claim to learn disentangled representations via unsupervised video decomposition.




In the rebuttal the authors have argued that their method studies a different problem from that of unsupervised video segmentation, and thus a comparison to those methods is unnecessary. Moreover, they mentioned that no established metrics exist for evaluating the degree of disentanglement of a representation, thus no new experiments can or should be added to the paper. I can see that there is a difference between image-based scene decomposition and unsupervised video segmentation. Once you move to video-based methods, however, the difference start to elude me. At least in the classical works, such as Brox and Malik, ECCV'10, the problem is defined in exactly the same way - decomposing a video into object/background regions in a fully unsupervised way. The only difference I can see is that in those works decomposition was the end goal, whereas this paper attempts to use it as a surrogate task for representation learning. This would be a valuable contribution if the authors could show that the resulting representations are superior to those learned with other unsupervised objectives (say, contrastive learning) at least for some tasks (say, object detection). Unfortunately, such evaluation is missing from the paper, thus I still find find that the benefits of the proposed approach are not convincingly demonstrated.

---

> ### Author Response · Authors · 2020-11-23
> **Response to Reviewer 4**
>
> We thank the reviewer for the valuable comments and hope to show the merits of our work below.
>
> **Contribution**
>
> There are a vast number of approaches for scene decomposition. In this work, we focus on solving it by decomposing a scene into a collection of components by grouping pixels within a probabilistic framework. Each component therefore is encoded into its own latent vector representation, which allows us to reconstruct individual objects independently from the rest of the scene (see Figure 1 (“decomposition”)), as opposed to the standard VAE framework, where the entire scene is encoded into a single vector representation. The goal is to make these latent representations encode only the information related to each individual object, and yet the information should be rich enough to have the reconstruction that matches the ground truth image. To evaluate the latter we employ traditional mean squared error (MSE). Because our model groups pixels, the outputs of the model are naturally individual masks corresponding to each object. If the model has separated the objects correctly, the produced mask will also have a high accuracy. Hence, to evaluate a proper separation of the objects it is natural to compare the produced masks with the ground truth masks, which suggests “mask segmentation” as a valid proxy for “object separation”. We would also like to point out that MSE and ARI are metrics that were used in our baselines models as well, meaning its use was necessary to ensure comparability of our evaluations.
>
> Despite extensive research in the static scenario, substantially less efforts have focused on scene decomposition in the temporal domain. The temporal aspect makes the task more challenging, since objects can leave or emerge into the frame, they can collide or occlude each other. At the same time, having additional temporal information, that can be explicitly incorporated into the probabilistic framework, can actually improve the model performance and help decompose objects more accurately. And this is exactly what we have demonstrated in our paper. We are showing that by modeling the dynamics we have obtained more precise separations and higher accuracy reconstructions of separate objects.
>
> Finally, addressing your point about the interpretable disentangled representation, which surely is an important and nice feature of the model. We have shown that by using the spatial broadcast decoded similarly to IODINE(Greff et al.) we have disentangled representations, and can efficiently interpolate in the latent space, by traversing space, size, color and other dimensions of each individual object (See Figure 12 in Appendix) and to our knowledge, there is no established quantitative metric for disentanglement. Please note that in order to see appearance change only in one of N objects, when traversing a certain dimension, you need this latent code to be reconstructed into a separated object and this is what we measure with the mask segmentations.
>
> **Minor comments response**
>
> 1. The issue with using the intersection of unions in evaluating segmentation quality is the inconsistent slot assignments within the same model across a sequence of frames. While our model tends to be very consistent in the object assignments, other methods like IODINE often reorder the slots (see Figure 11) requiring a non-trivial logic to figure out the correct mask to compare to and possibly wrong results.
> 2. When evaluating our model on the CLEVRER dataset in Table 1, we used 6 object slots (K) for scenes containing 3-5 objects. While for the generalization experiments reported in Table 2, we used 7 object slots with 6 objects per scene. The expected number of objects is closer to the actual number of objects in the scene in the second experiment, leading to less redundant slots and thus higher accuracy.
> 3. Thank you for the suggestion, since Reviewer #1 also pointed out this reference, we have added a discussion of the comparison as a general reply and have added it to the paper as well.

---

> > ### Comment · AnonReviewer4 · 2020-11-25
> > **Response**
> >
> > I thank the authors for their thorough response. However, I still find that an appropriate experimental evaluation is missing in this work. If the main task is in fact scene decomposition, then the method needs to be compared to computer vision approaches for unsupervised video segmentation listed in my original review. If the task is representation learning, the value of the learned representations has to be demonstrated (say, by fine-tuning them on some downstream task). If the goal is learning disentangled representation, and no quantitative measures of disentanglement exist, they need to be introduced. If introducing such measures is impossible for some reason, the author should look at different problems.

---

> > > ### Author Response · Authors · 2020-11-25
> > > **Response to Reviewer 4**
> > >
> > > We thank the reviewer for their prompt reply and would like to address the raised concern.
> > >
> > > There are many paper on scene decomposition problem, which makes it a field on its own. Scene decomposition is different task from unsupervised video segmentation or disentangled representation. Moreover, many of the papers some of which were our baselines use exactly the same protocols of evaluation. Here are some of them:
> > >
> > >   [1] Locatello, Francesco, et al. "Object-centric learning with slot attention." Advances in Neural Information Processing Systems 33 (2020).
> > >   [2] Greff, Klaus, et al. "Multi-object representation learning with iterative variational inference." Proceedings of the 36th International Conference on Machine Learning (2019)
> > >   [3] Van Steenkiste, Sjoerd, et al. "Relational neural expectation maximization: Unsupervised discovery of objects and their interactions." International Conference on Learning Representations. (2018)
> > >   [4] Greff, Klaus, Sjoerd Van Steenkiste, and Jürgen Schmidhuber. "Neural expectation maximization." Advances in Neural Information Processing Systems. (2017)

---

### Official Review · AnonReviewer3 · 2020-10-23
**Novelty and contribution should be clarified**

**Rating:** 6
**Confidence:** 3

**Review:**

In this paper, the authors propose to better explicitly utilize the sequential information in the video to improve the performance of unsupervised scene decomposition in video. Concretely, 2D LSTM is used to combine the advantages of iterative inference and temporal information. By appropriately using the inferred results in the previous time step, the number of interactive inference steps is decreasing, which finally results in the O(R^2+T) complexity.

The main concern about this paper is the novelty. First, unsupervised video decomposition, like Sqair and R-NEM all use the temporal information, and the iterative inference method has been used in previous work like IODINE. It seems that this paper extends the IODINE to the video setting with a modification that decreasing the iterative steps as time increases. The authors should better illustrate their novelty and contributions of this paper.

The presentation is generally clear in this paper. However, the figures can be improved. First, figure 1 seems meaningless. I don’t the relationship between the top row and the bottom row in the top panel of figure 1. The captions said about the emergence of new objects, but I cannot find them in the figure. Highlight them with some red bounding boxes is helpful. Second, panel (b) of Figure 2 is really chaotic. I don’t know what’s the main purpose of this panel. Again, highlight the part you want to emphasize.

Finally, there are some related but not discussed work:
[a] Chen, Mickaël, Thierry Artières, and Ludovic Denoyer. "Unsupervised object segmentation by redrawing." Advances in Neural Information Processing Systems. 2019.
[b] Arandjelović, Relja, and Andrew Zisserman. "Object discovery with a copy-pasting gan." arXiv preprint arXiv:1905.11369 (2019).
[c] Xu, Taufik, et al. "Multi-objects generation with amortized structural regularization." Advances in Neural Information Processing Systems. 2019.

---

> ### Author Response · Authors · 2020-11-23
> **Response to Reviewer 3**
>
> We thank the reviewer for valuable feedback and answer raised questions below.
>
> 1. Paper contribution.
> Our method allows an effective use of temporal information in object-centric decompositions of colored video data. This places our approach between methods like R-NEM, which strictly operates on binary data, and IODINE, whose usage of temporal information is ad-hoc and produces results of limited quality (Table 1). In practice, we leverage a 2D-LSTM but also employ an implicit modeling of dynamics by incorporating the hidden states into a conditional prior. For further details, please refer to the response to R1. We agree that these contributions could have been expressed more clearly and have revised the corresponding sections in the paper.
>
> 2. Figure clarity
> We have revised Figure 1. We will work on revising Figure 2 for the camera ready version. We would welcome any specific ideas on how readability of the figure(s) can be further improved and will follow reviewer guidance and suggestions.
>
>
> 3. Related work
> We thank the reviewer for pointing us to relevant references; we have included the references in the related work section:
>
>   A more recent variant of AIR, proposed by [c], captures dependencies between objects through posterior regularization. Similar to other attention-based methods, this approach is limited to bounding box detections and evaluated on scenes without perspective distortion.
> ReDO [a] uses a GAN-based model built around the assumption that object regions are independent, guiding the generator by drawing objects’ pixel regions separately and composing them after segmentation. [b] employs the same principles but guide the generator by copying a region of an image into another one. Both architectures are shown to operate on static images only (no perspective distortion for [b]) and do not have a clearly interpretable latent space or prediction capabilities.

---

### Official Review · AnonReviewer1 · 2020-10-28
**A well-executed application of iterative inference to the video domain**

**Rating:** 7
**Confidence:** 4

**Review:**

The paper presents a model for the unsupervised decomposition of videos into objects. It builds on
previous models operating on individual images such as IODINE and GENESIS, and extends them to the
video domain by conducting iterative inference also across the time dimension, leading to a
computation graph resembling a 2D grid. It is shown that the proposed model provides more accurate
scene decompositions than previous ones, and generalizes better to different numbers of objects.
Predictions of future frames appear somewhat less accurate.


Strengths:
 1. The proposed method closes an important gap in the landscape of unsupervised object-based
    models: It is the first such method to decompose colored videos using pixel-wise segmentation
    masks. The method is well motivated and clearly presented.
 2. The experimental evaluation follows the precedent set by R-NEM, IODINE, and others. It appears
    to have been rigorously conducted and shows encouraging results in terms of scene decomposition
    and generalization. The reduced reliance on color information, which has plagued previous models
    such as MoNET, is especially welcome.
 3. The main technical contribution, the two-dimensional refinement method, seems to provide clear
    benefits over previous ad-hoc implementations such as SEQ-IODINE, both in terms of runtime and
    performance.

Weaknesses:
 1. As noted by the authors, predictive performance is somewhat lacking, which may be explained by
    the lack of relational components.
 2. Using the inference network as opposed to the generative model for prediction seems to run
    counter to the idea of learning a generative model to accurately model the distribution of the
    data. In particular, because the generative distribution $p(z_t | x_{<t}, z_{<t})$ seems ill-equipped
    to model object interactions.
 3. Like all unsupervised object-models, this work is still quite far away from working on natural
    videos.

Overall, I believe this paper fills an important gap that many members of the community were waiting
to see filled. It is well executed and adds some interesting ideas to the toolset of unsupervised
object models. I therefore recommend acceptance.

Questions:
 1. It seems that during inference, interactions between slots may be taken into account through the
    gradients of the likelihood. But during prediction, those are set to zero, and the prior
    over $z_t$ seems to operate slot-wise as well. So can object interactions be computed at all
    during prediction, or are the slots predicted independently of one another?
 2. The choice of $\hat{R}$ seems to express the idea that once a good latent state has been inferred,
    a single refinement step is enough to update it given new observations. Would additional steps
    be beneficial when receiving unexpected input, such as frames with newly appearing objects on
    CLEVRER? Could the number of steps perhaps even be selected dynamically based on e.g. the
    reconstruction error?

---

> ### Author Response · Authors · 2020-11-23
> **Response to Reviewer 1**
>
> We thank the reviewer for highlighting the contribution and strengths of the paper and the overall positive view. We address the questions below.
>
> 1. As you correctly stated the slots relations are only taken into account in the final step of the generation where the likelihood is computed. When the gradients are propagated back through the network the interaction and relative positions of the objects are learned by the network. Therefore the answer to your question is yes, the slots become predicted independently, when the gradients are set to zero. This may explain the fast decline of the prediction accuracy as you stated. However, prediction is not the main goal of our work, and we were focusing on improved accuracy of the decompositions. By showing that the prediction works for a few frames we demonstrated that in our framework one can get short predictions for “free”. Review also mentioned the “relational component”, which we have considered, however in the architecture of our model adding a slot self attention would further increase the complexity, and thus significantly hurt its scalability. In Section E of the Appendix we have mentioned adding a relational component in the future work, however, we should be mindful of making this as scalable as possible, so making a relational component scalable within these types of models. We will consider this in future work, but view this as being outside the scope of the current paper.
>
> 2.  The quality of the segmentation does not particularly degrade when a new object enters the scene (see Figure 13), especially compared to baselines.
> However, it is a good suggestion to select the number of refinement steps dynamically as it would make the framework more flexible. In general, this would lead to an increase in the number of refinement steps “on average” compared to the current model, which, in preliminary experiments, we saw to improve the performance. We will experiment with this more thoroughly and include a qualitative example in the paper.

---

### Official Review · AnonReviewer2 · 2020-10-29
**An interesting paper with insufficient experiments**

**Rating:** 6
**Confidence:** 4

**Review:**

This paper presents a new model for unsupervised video decomposition based on the multi-object scene representation of IODINE. Basically, it makes use of 2D-LSTMs to provide the IODINE framework with a better ability to capture temporal dependencies.

1.  My first concern is about the novelty of the proposed model. The most important contribution, if I understand it correctly, is the use of 2D-LSTMs in the iterative amortized inference for temporal modeling. It does extend IODINE into the spatiotemporal context, but it only provides limited new insight into this research field.

2. Another concern is about the CLEVRER experiments. From Table 2, a counter-intuitive observation is that IODINE significantly outperforms Seq-IODINE. There might be two reasons: (a) In this dataset, the temporal dependencies between frames are relatively weak, because future frames depend not only on previous ones but also on a series of actions. (b) Seq-IODINE may not be a strong baseline model; If so, the authors might include other existing approaches specifically designed for video data, such as R-NEM and DDPAE [Hsieh et al., 2018].
[Hsieh et al., 2018] Learning to Decompose and Disentangle Representations for Video Prediction.

3. Although I am aware that R-NEM and IODINE were also evaluated on the synthetic datasets only, I recommend that the authors validate the model on real-world datasets of non-rigid objects and more complex structures.

4. If the authors can provide a direct qualitative comparison with R-NEM and IODINE in Figures 3-4, it will be easier to understand the advantages of the proposed model.

5. In Section 2, the authors could use a separate paragraph to describe the main differences between the proposed model and previous ones, in particular R-NEM and IODINE.

---

> ### Author Response · Authors · 2020-11-23
> **Response to Reviewer 2**
>
> We thank the reviewer for their thoughtful feedback and valuable suggestions. We address the comments below.
>
> 1. It is correct that the use of a 2D-LSTM within the iterative amortized inference framework is one of our main contributions. We emphasize that the extension from the 1D case is non-trivial and the use of conditional priors, in particular, proved to be an effective modeling choice in the inference process, as illustrated by our ablation study (Table 3). As a direct consequence of this contribution, we obtain:
>     - A fast and efficient scene decomposition with interpretable and disentangled latent representation, which is preserved over the whole sequence.
>     - Implicit probabilistic dynamics for each object in the scene, resulting in higher overall accuracy and explicit temporal hidden states.
>
>   Furthermore, while we present our architecture in the context of video decomposition, we believe it to be valuable in a variety of other scenarios requiring temporal consistency and hence view it as having a much broader appeal than this application may suggest.
>
>   Finally, we want to mention that we have extended the CLEVRER dataset and have made it publicly available to the community.
>
> 2.
>     - Apart from suffering from exploding gradients and lack of explicit dynamics (see Section 4.3), SEQ-IODINE (unlike IODINE) does not include iterative refinements of each frame, which explains why it performs worse. This also leads to the first frame of a sequence being worse than the rest, which additionally lowers the accuracy of the decompositions.
>     - R-NEM and DDPAE are both tailored to binary data. Additionally, DDPAE is based on an attention mechanism and, as such, does not approach scene decomposition by grouping pixels. Consequently, DDPAE is not evaluated on segmentation tasks, as it cannot generate masks. The paper also focuses on physics modelling, where it is a strong baseline for the prediction of dynamics (see below).
>
>   In order to compare our model to R-NEM and DDPAE on CLEVRER, we would need to binarize it. However, that would render occlusions, which are a valuable component of the 3D scenes in CLEVRER, meaningless. We have therefore decided not to include these comparisons.
>
>   Nonetheless, we agree that a discussion of DDPAE is important and have added a paragraph to our related work section. Additionally, we conducted an experiment comparing the predictive power of our model to DDPAE on the Bouncing Balls dataset (Figure 4). We measure performance as the cosine similarity of the velocity prediction of both models. As expected, our model outperforms DDPAE on the first three frames and then declines in quality. This behavior is not surprising and in line with results reported in the paper (see Figure 5). Simulation is not our model’s main use case, though, and should be seen as an auxiliary task based on our model’s object dynamics.
>
> 3. We agree that evaluating our method on a real-world dataset would make the paper more well-rounded. For this reason, we are including an evaluation on the Grand Central Station dataset (Figure 7), which is the only real-world dataset used to evaluate models similar to ours. In order to keep the number of objects in the scene manageable, we have extracted a cropped fragment from the video sequence. We have added dataset details as well as a qualitative evaluation to the paper. Since the dataset does not contain ground truth segmentation masks, a quantitative evaluation was not possible. We think running  (SEQ-)IODINE on the same dataset will give more meaning to the experiment and will do it for the final version of the paper.
>
> 4. While the limited rebuttal period did not allow us to generate these results for R-NEM, we agree that such a direct qualitative comparison would highlight the advantages of our method and will include it in the final version of the paper. For a comparison with IODINE and SEQ_IODINE, please see Figure 13 in the Appendix.
>
> 5. We have included this discussion in the paper.

---

### Author Response · Authors · 2020-11-23
**General response to the reviews and list of major modifications**

We thank all reviewers for their valuable comments and constructive feedback. We have revised the paper with the following changes.

- Added a reference to a more recent variant of AIR pointed out by R3.
- Included a discussion of [Hsieh et al., 2018][Arandjelović et al., 2019][Chen et al., 2019] to Section 2.
- Improved clarity of Figure 1 through highlights.
- Clarified the differences of our method from existing research at the end of Section 2.
- Added quantitative evaluation to DDPAE  (Figure 4) and discussion in the predictions experiments (Section 4).
- Included qualitative evaluation on a real-world dataset (Figure 7).

---

### Decision · Program_Chairs · 2021-01-07
**Final Decision**

**Decision:**

Reject

**Comment:**

The reviewers appreciate the spatio-temporal formulation of amortised iterative inference.
However, the paper does not clearly state what is the end goal: if the end goal is video object segmentation, it should compared against other unsupervised object segmentation methods. If the goal is representation learning, it should evaluate the merit of the recovered representations, e.g. by fine-tuning them on some downstream task.